# DMLAR: Distributed Machine Learning-Based Anti-Collision Algorithm for RFID Readers in the Internet of Things

**Rachid Mafamane** [1] , **Mourad Ouadou** [1,*] , **Hajar Sahbani** [1,2] , **Nisrine Ibadah** [1] and **Khalid Minaoui** [1]

1 LRIT Laboratory, Faculty of Science, Mohammed V University, Rabat 1014, Morocco; rachid_mafamane@um5.ac.ma (R.M.); h.sahbani@emsi.ma (H.S.); nisrine.ibadah@gmail.com (N.I.); khalid.minaoui@fsr.um5.ac.ma (K.M.)
2 SmartiLab Laboratory, Moroccan School of Engineering Sciences (EMSI), Rabat 10000, Morocco
* Correspondence: mr.ouadou@um5r.ac.ma

**Abstract:** Radio Frequency Identification (RFID) is considered as one of the most widely used wireless identification technologies in the Internet of Things. Many application areas require a dense RFID network for efficient deployment and coverage, which causes interference between RFID tags and readers, and reduces the performance of the RFID system. Therefore, communication resource management is required to avoid such problems. In this paper, we propose an anti-collision protocol based on feed-forward Artificial Neural Network methodology for distributed learning between RFID readers to predict collisions and ensure efficient resource allocation (DMLAR) by considering the mobility of tags and readers. The evaluation of our anti-collision protocol is performed for different mobility scenarios in healthcare where the collected data are critical and must respect the terms of throughput, delay, overload, integrity and energy. The dataset created and distributed by the readers allows an efficient learning process and, therefore, a high collision detection to increase throughput and minimize data loss. In the application phase, the readers do not need to exchange control packets with each other to control the resource allocation, which avoids network overload and communication delay. Simulation results show the robustness and effectiveness of the anti-collision protocol by the number of readers and resources used. The model used allows a large number of readers to use the most suitable frequency and time resources for simultaneous and successful tag interrogation.

**Keywords:** RFID; IoT; machine learning; collision; MAC layer; wireless sensor network

## 1. Introduction

Radio Frequency Identification (RFID) is a technology that allows the identification of fixed and mobile objects via wireless communication. It is characterized by a few main features such as reading rate, data storage capacity, and radio frequency energy harvesting. RFID technology is also differentiated in terms of frequency band (LF, HF, UHF) and type of tag (passive or active). The evolution of technology and the miniaturization of electronic components make RFID particularly suitable for data acquisition by tag-integrated sensors. An RFID system consists of two main entities (Figure 1): tags [1] and readers [2]. The main component of a RFID system is the reader, which operates as follows:

- It supplies energy to the tag via radio waves;
- It interrogates the tags in its reading field;
- It receives and forwards responses from the tags to the corresponding applications.

RFID readers can perform in different forms depending on the requirements of the proposed application. Thus, readers may be fixed (e.g., supermarket checkout, luggage control in airports, supply chain) or mobile (e.g., robotics, healthcare).

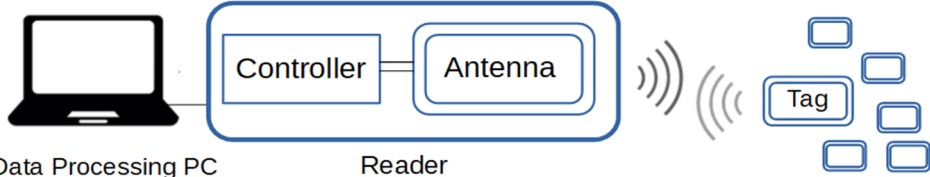

**Figure 1.** RFID system communication process.

In RFID systems, the frequency bands used are: 125–134 kHz for low frequency (LF); 13.56 MHz for high frequency (HF); and 860–960 MHz and 2.4–2.45 GHz for ultra-high frequency (UHF). The main difference between the frequency range is the type of coupling used for communication and tag powering. The RFID reader's power combined with the appropriate antenna defines several optimal readings ranges: proximity RFID readers (up to 25 cm); neighbor RFID readers (up to 1 m); medium-range readers (up to 9 m); and long-range readers (several hundred meters).

Application domains of RFID systems generally include monitoring, control and supervision aspects [3]. Some of the areas using such systems include healthcare [4], smart warehouses [5], indoor tracking [6], brain research experiments [7], modern agriculture [8], and supply chain management [9–13]. In healthcare, for instance, patients are being equipped with RFID combined with sensor tags in order to receive real time medical data.

Hence, the deployment of the RFID network in healthcare offers many advantages and brings new comfort to patients. As a generalization, the Internet of Things (IoT) [14] includes a large number of interconnected devices using various protocols to communicate. The IoT model consists of different layers: sensing, access, network and application. RFID represents a powerful communication technology in IoT since it allows identification with physical addresses to connect the device to the IoT [15,16].

However, RFID systems suffer from several problems related to the random deployment of tags or readers. Such deployment can cause two major problems, namely, collisions and data redundancy. In this paper, we focus on the collision problem due to random deployment, mobility and same-time multiple access. We consider, in this work, tags combined with sensors for a wireless RFID sensor network [17–22] and we take healthcare IoT as an application because of the high mobility of tags and readers in such a system [23].

In this context, reader collisions and interference can be classified into two categories [24]: Reader-to-Reader Interference (RRI) and Reader-to-Tag Interference (RTI). RRI collision, represented in Figure 2a, occurs when two readers are located in the same interference range using the same data channel and simultaneously interrogate the tag. However, RTI collision (Figure 2b) occurs when readers attempt to simultaneously interrogate the same tags located in their reading range regardless of the frequency used. Several researchers have proposed their solutions to such collision problems [25–34]. In our case, we bring novelty by:

- Implementing an Artificial Neural Network for the mobile reader RFID anti-collision MAC layer protocol for the first time;
- Using and exploiting the capacity of different models (RRI, RTI and INT) and their readers for collision detection.

The remainder of this paper is organized as follows: Section 2 presents the related works for RFID collision avoidance protocols. Our machine learning-based algorithm is described in Section 3. The simulation results are discussed in Section 4. Finally, Section 5 gives a conclusion and perspectives for this work.

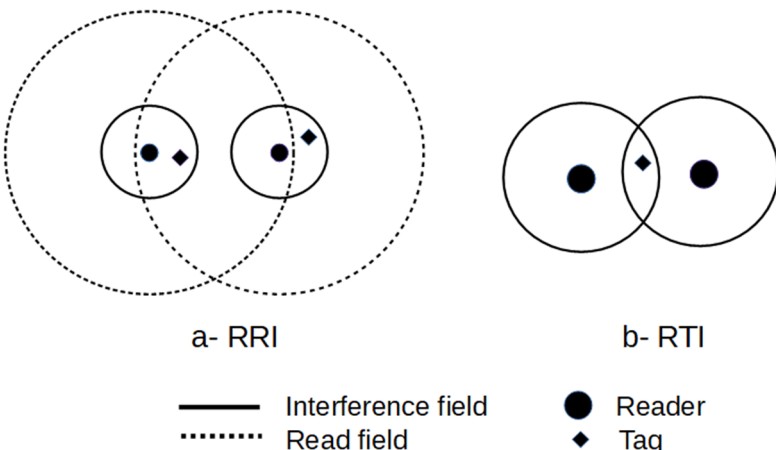

**Figure 2.** RFID collisions. (**a**) Reader-to-Reader Interference; (**b**) Reader-to-Tag Interference.

## 2. Related Work

To solve the collision problem of RFID readers, several researchers have proposed solutions for the MAC layer. In this section, we mention different protocols from the literature according to architecture (centralized or distributed) and to the learning system used to control the access to the channel (Neural Network or Immune Network) (Figure 3).

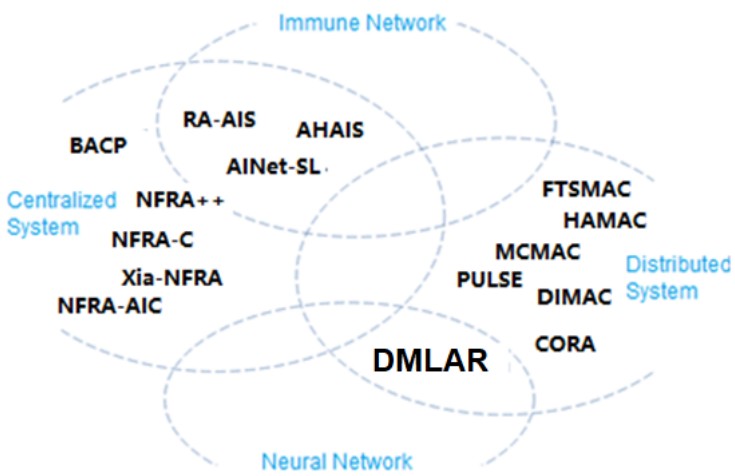

**Figure 3.** RFID protocol classification.

Pulse [35], is a distributed anti-collision protocol that uses a single data channel for tag interrogation and a control channel to broadcast beacon messages to the neighboring reader when tags are currently reading to avoid channel collisions. This allows many RFID readers on the network to be disabled, which degrades system performance.

MCMAC [36] (Multi-Channel MAC protocol) is an RFID anti-collision protocol that uses multiple data channels for tag interrogation and a single control channel for notification exchange between readers to avoid interference on the data channel. Access to these channels is controlled by a random algorithm in which the reader listens to the control channel to analyze the control messages if there are free channels for tag interrogation. The simultaneous reading of a tag by different readers causes RTI collisions even if they use different frequencies because the control channel can only resolve RRI collision.

CORA [37] (Coverage-Oriented Reader Anti-Collision) is a protocol for mobile and time-critical RFID readers. Each reader in the RFID network must identify the neighbors in collision by learning the state of its local environment in order identify whether to activate or not. For this, each reader informs its neighbors in the interference field of the TS used. This allows the reader to calculate the number of readers in collision (same TS) and in

non-collision (different TS) according to the used TS. Thus, the reader can interrogate tags when the number of readers in non-collision is higher than the readers in collision.

FTSMAC [38] is a distributed hybrid protocol based on TDMA and FDMA. This algorithm makes it possible to create efficient frequency and time slot distribution schemes. These schemes allow the integration and activation of a maximum number of RFID readers to cover most of the tags.

DiMCA [39] (Distributed Multi-Channel Collision Avoidance) is a CSMA-based protocol that avoids Reader-to-Reader and Reader-to-Tag interference by using a notification system allowing each RFID reader to declare or release data channel occupation. The control message can reach the college readers in RRI via the first control channel and the readers in RTI via a second control channel. This protocol minimizes interference from data channel usage, but is still unable to avoid control channel collisions. The message overhead generated by the readers affects the communication delay.

HAMAC [40] (High Adaptive MAC) is a dedicated protocol for mobile and large-scale RFID networks. Based on CSMA, this protocol is used to reach a maximum backoff value to avoid more collisions in the WSN. The readers' RFID systems do not require additional resources or components to improve performance. The reader dynamically controls its window of contention according to network congestion on the available frequencies by linearly or multiplicatively decreasing.

BACP [41] (Beacon Analysis-Based RFID Reader Anti-Collision Protocol) is an algorithm based on TDMA and FDMA channel access techniques. This protocol is centralized, the server signals the start of the round, and each RFID reader must wait for the reception with the control message containing the priority code to make their decision. Querying tags do not require readers to broadcast beacon messages to the neighborhood.

NFRA [42] (Neighbor Friendly Reader Anticollision) is a popular RFID anti-collision protocol that is centralized and powerful. Mobile readers are managed by the polling server that divides the time over several samples. The server exchanges certain control messages with the readers regarding time slots to distribute the TSs properly, avoiding RTI and RRI. Following the NFRA protocol, such variants are proposed by modifying the contention procedures NFRA++, NFRA-C, and Xia-NFRA.

NFRA++ [43]: The reader using this protocol updates its priority level at each waiting round after receiving a command from the server. The number of waiting rounds is estimated according to the previous status of readers.

NFRA-C [44] is effective for dense networks and uses a counter to store the successful communication logs of each reader. The reader broadcast its countered via a beacon to inform the neighbors of the preset collisions.

Xia-NFRA [45] is suitable for dense and mobile RFID networks. To achieve this goal, the algorithm allows the reader to use a new subframe within the NFRA protocol transmission frame. To improve transmission accuracy an improvement in the backoff is implemented.

NFRA-AIC [46] (RFID Reader Anti-Collision Protocol with Adaptive Interrogation Capacity) is the extension of the NFRA centralized anti-collision algorithm family. RFID readers determine the time and duration of reading the tag according to the number of tags located in their reading field.

RA-AIS [47]: To solve the Reader-to-Reader Collision Avoidance Model (R2RCAM), an artificial Immune Network is proposed for the optimization of resource allocation. The candidate antibody corresponds to the optimal frequential and temporal resources. The antigen representing the resources allocated for the R2RCAM model and the mutation phase correspond to the effective dynamic adjustment of the readers' interrogation field.

The AHAIS [48] (Adaptive Hierarchical Artificial Immune System) increases the convergence rate and the efficiency of the RFID system. In this algorithm, the antibodies are classified in two levels: a common swarm (CS) with lower affinity at the top-level and an elitist swarm (ES) with a higher affinity at the bottom-level. In the mutation operator,

the ES antibody uses self-learning and local research and the CS antibody emphasizes ES learning and global research.

AINet-SL [49] is an Artificial Immune Network with social learning, inspired from the social behavior of animals. Similar to AHAIS, this algorithm also classifies antibodies into two swarms: ES and CS. Both successively undergo the self-learning and social-learning mutation strategy. The mutation phase uses two learning strategies: stochastic social learning (SSL) and heuristic social learning (HSL).

## 3. Collision Detection Algorithm Based on Machine Learning

In this paper, we introduce a Neural Network model adapted for a RFID wireless sensor network to avoid collisions.

Firstly, it was necessary to determine the appropriate learning method for our RFID network. For our case, we chose the Artificial Neural Network (ANN) method adapted to our context. In fact, as an extended logistic regression system, the ANN incorporates extra layers of feature combinations. With these additional layers, we can learn more and obtain better results. This choice was also consolidated by the results we obtained by comparing the ANN, Logistic Regression and Decision Tree models in terms of collision prediction in the simulation section (see Section 4.3)). The results obtained showed that the ANN model was the most adequate in terms of prediction reliability compared to the other models.

We chose the ANN using two collision models (RRI and RTI) instead of the multiclass RRI–RTI model (with four outputs), based on the performance comparison (see Table 1).

**Table 1.** Performance comparison.

| ANN Model | RRI–RTI | RRI | RTI |
|---|---|---|---|
| Performance | 0.21 | 0.85 | 1.74 |

We operated varying parameters during simulations as inputs for the ANN of each RFID reader and used collision prediction as an output. The hidden layer consisted of a range of constraints, which included: available resources (frequency and time slots), real-time requirements of WBAN, environmental factors, device factors, energy factors, etc.

Thus, we used the Artificial Neural Network as a learning method to resolve RRI and RTI collisions. The dataset was used to learn and train a suitable model for different deployment and movement scenarios. Typically, a Neural Network consists of three layers of interconnected neurons:

- The input layer introduces the data collected by the reader in each transaction;
- The hidden layers are the core of our concept where the relationships between the variables are highlighted;
- The output layer predicts the presence or absence of each type of collision.

The objective of our proposed methodology described in Figure 4 is as follows:

A dataset is created and shared by each reader that will then broadcast a learning model on the network to select the model with the best prediction ratio of RRI and RTI collisions (see Section 4).

### 3.1. Collection Phase

3.1.1. Construction of Dataset

Before starting learning, we set different scenarios where each reader collected and identified the presence of collisions at each period (Algorithm 1). Thus, the readers constructed their dataset in two steps:

- During the dedicated simulation for the collection phase, each reader adds an entry (Equations (2) and (3)) in its dataset at each period;
- Readers broadcast their final dataset to all readers to increase their dataset.

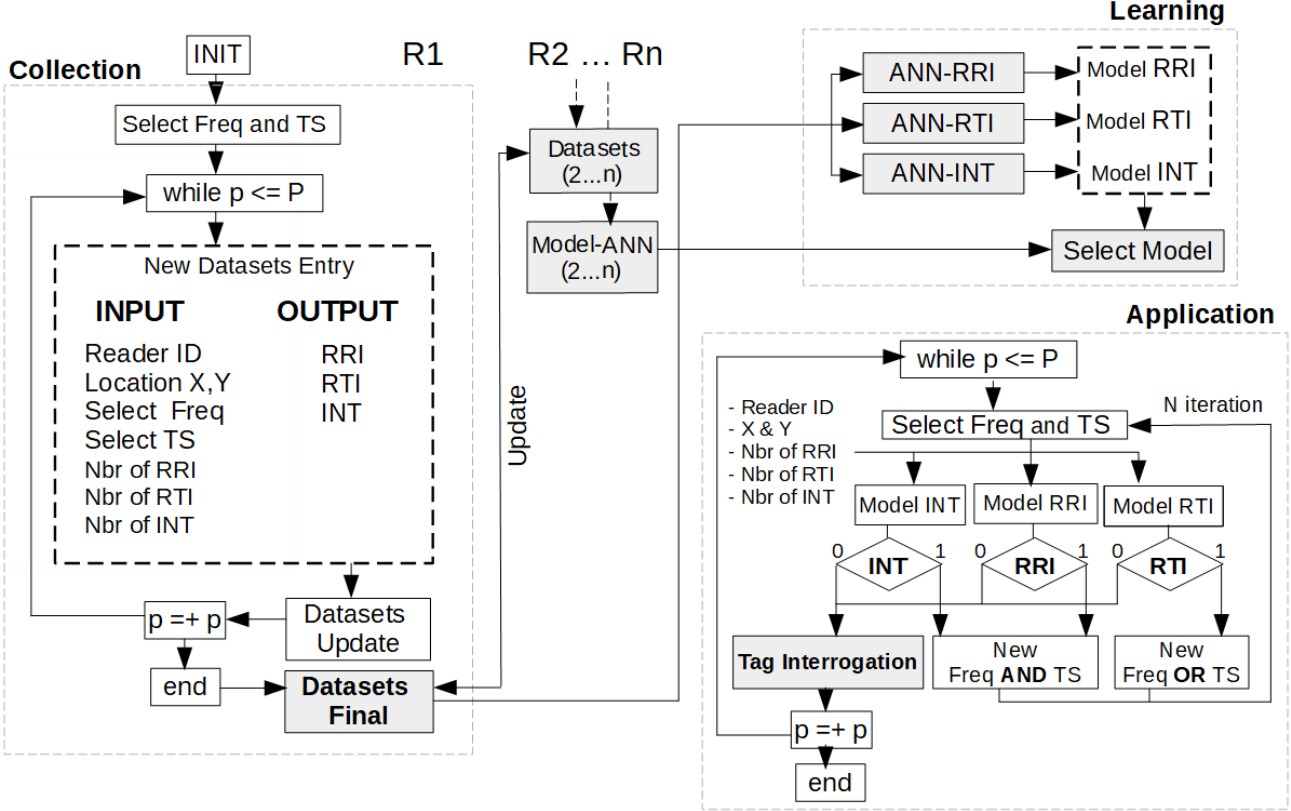

**Figure 4.** Proposed algorithm scheme.

Thus, in the collection phase (Figure 4), each reader *Rx* prepares its dataset throughout the simulation. Readers update their datasets *Dx* at each movement represented by an instance of periods '*p*'. Therefore, the number of data entries will be equal to the number of simulation periods. Following the simulation, the readers {*R1 . . . Rn*} broadcast and update their datasets. The distribution process is described in the algorithm (Algorithm 2). Thus, the dataset size becomes:

$$dsatasetSize = nbrR \times p \qquad (1)$$

where *nbrR* is the number of readers and p is the number of reader movement steps.
A dataset entry consists of two parts: input and output:

$$datasetInput(Rx, pi) = [ReaderID, \ Location, \ NewFreq, \ NewTS, \ NbrRRI, \ NbrRTI, \ NbrINT] \qquad (2)$$

where:

- *Nbr RRI*: number of readers in the Reader-to-Reader Interference domain;
- *Nbr RTI*: number of readers in Reader-to-Tag Interference domain;
- *Nbr INT*: number of readers in the Interference domain.

$$datasetOutput(Rx, pi) = [RRI, \ RTI, \ INT] \qquad (3)$$

where:

- *RRI*: Boolean represents the presence or absence of Reader-to-Reader Interference;
- *RTI*: Boolean represents the presence or absence of Reader-to-Tag Interference;
- *INT*: Boolean represents the presence or absence of Reader-to-Reader or/and Reader-to-Tag Interference.

At the end of this phase, all readers have sufficient data to perform the learning in the next phase.

---

**Algorithm 1:** Number of interfering readers and interference detection.

---

*CCP*//*control channel power*
*receiveP*//*receive power*
*MsgD = [senderID]*//*discovery message*
*Freq = random[freq1 ... freqN]*//*select a random frequency*
*TS = random[ts1 ... tsN]*//*select a random Time Slot*
*MsgR = [receiverID, Freq, TS]*//*response message*
*RRI, RTI, INT = 0*//*Interference detection*
*Step1—————————*
*Wait (Backoff)*//*wait for a random backoff to avoid collisions on the control channel*
*Rx broadcast MsgD*//*each reader broadcast a MsgD message to detect readers in the control field*
*if receive = MsgD*//*receiving the MsgD message*
*send MsgR*//*reply to the sender reader MsgR message*
*elseif receive = MsgR*//*receiving the MsgR message*
*Step2—————————*
*//———RRI detection———-*
*if receiveP(MsgR) < CCP*//*received signal power less than control channel power*
*nbrRRI = nbrRRI + 1*//*new reader in RRI*
*if MsgR.Freq = this.Freq*//*RRI reader uses the same frequency*
*RRI = 1*//*RRI collision detection*
*//——— RTI detection———*
*elseif receiveP(MsgR) > CCP*
*nbrRTI = nbrRTI + 1*//*new reader in RTI*
*if MsgR.Freq = this.Freq and MsgR.TS = this.TS*//*RRI reader uses the same frequency and TS*
*RTI = 1*//*RTI collision detection*
*end*
*end*
*//——INT detection——*
*nbrINT = nbrRRI + nbrRTI*//*new reader in INT*
*if RRI=1 or RTI=1*//*RRI or RTI collision exist*
*INT = 1*//*INT collision detection*
*else*
*INT = 0*//*No INT collision*

---

---

**Algorithm 2:** Dataset and model broadcast.

---

*Dx*//*Rx reader dataset*
*MxRRI, MxRTI, MxINT*//*Rx reader RRI, RTI and INT model*
*nbrReader*//*number of readers who received MsgM to avoid the infinite broadcast loop*
*readerID*//*reader identification*
*MsgM = [readerID, nbrReader, Dx, MxRRI, MxRTI, MxINT]*//*model message*
*readerSet*//*reader list*
*Reader.size*//*number of readers in the simulation network*
*Wait (Backoff)*//*wait for a random backoff to avoid collisions at the control canal*
*Rx broadcast MsgM*//*each reader broadcast a MsgM in the control canal*
*if receive = MsgM*//*receiving the MsgM message*
*while MsgM.nbrReader < Reader.size*//*while the network readers not all received* MsgM *message*
*if MsgM.readerID not existe readerSet*//*MsgM from reader ID is not received*
*update readerSet*//*add the new reader to reader list*
*MsgM.nbrReader = MsgM.nbrReader +* 1//*a new reader receives MsgM message*
*broadcast MsgM*//*broadcast a MsgM message at the control canal*
*end*
*end*

---

### 3.1.2. Number of Interfering Readers and Interference Detection

In order to collect the external information for the input (*NbrRRI*, *NbrRTI* and *NbrINT*) and output (*RRI*, RTI and *INT*) of the dataset, a notification system (Algorithm 1) is developed. At each simulation period, all readers in the network wait for *Backoff* [38]. The

first reader to wake up *Ri* listens to the control channel *CC* during a period *T* in order to avoid collisions on this channel [38]. Then, reader *Ri* broadcasts a *MsgD* within its data channel interference field equal to the control channel reading field. The readers in this field reply with the *MsgR* frame to indicate the used resources. At this stage, the reader Ri compares the received power of the frames to calculate the number of readers that are possible to be in *RRI* (*receiveP* < *CCP*) or in *RTI* (*receiveP* > *CCP*). Thus, *Ri* will check if there is at least one collision of *RRI* (*MsgR.Freq = Ri.Freq*) or *RTI* (*MsgR.Freq = Ri.Freq and MsgR.TS = Ri.TS*). (Algorithm 1).

### 3.1.3. Readers' Broadcasting Dataset Process

In order to share datasets without collisions, we use an effective communication strategy after the collection phase. The reader *Ri* that wakes up first (*Backoff* = 0) [38] starts to broadcast the *MsgM* containing its dataset (*Di*). This message is received by the readers in the control channel reading field of *Ri*.

Each reader will compare the number of readers received in *MsgM* (*MsgM.nbrReader*) with the total number of readers in order to avoid a broadcasting loop. Then, it will forward the new *MsgM* to the next neighbors until all readers update their dataset. (Algorithm 2).

### 3.2. Learning Phase

During this phase (Figure 4), readers use the Artificial Neural Network (ANN) [50] to perform learning. Each reader sets three different ANN models, each one composed of an input of 7 neurons, 1 output of a single neuron and 3 hidden layers, constituted successively by 7, 10 and 3 neurons (Figure 5).

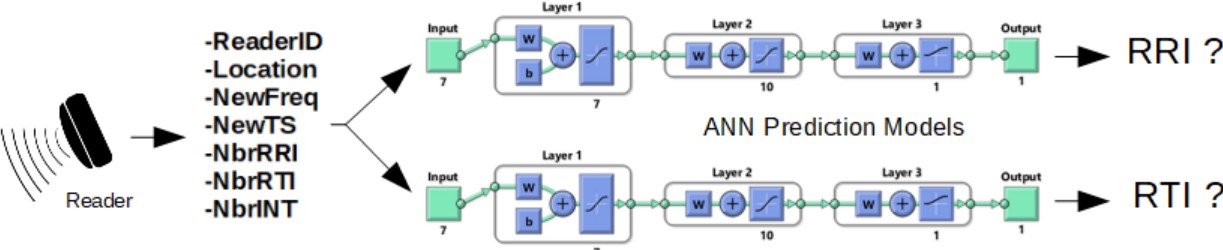

**Figure 5.** Artificial Neural Network architecture for RFID readers.

The three models used by readers are:

- MxRRI: RRI prediction model use Reader ID, Location, New Freq and Nbr RRI as input;
- MxRTI: RTI prediction model use Reader ID, Location, New Freq, New TS and Nbr RTI as input;
- MxINT: Interference prediction model use Reader ID, Location, New Freq, New TS and Nbr INT as input.

Following the learning phase, each reader broadcasts its ANN models (*RRI, RTI, INT*) and performance to its neighbors. For that, using the communication process (Algorithm 2), readers use the control channel *CC* to distribute the *MsgM* frame, activating in this case the three fields of the ANN models. The best ones are then selected. Thus, all readers will obtain the following set of models:

$$receivedModel = [\{M1RRI, M1RTI, M1INT\} \ldots \{MiRRI, MiRTI, MiINT\} \ldots \{MnRRI, MnRTI, MnINT\}] \tag{4}$$

where *i* is the ith reader and *n* is the number of readers in the network.

In summary, the objective is to exploit the diversity of the reader's learning experience in order to select the most suitable models for a given scenario.

### 3.3. Application Phase

In the last phase of this process (Figure 4: Application), all readers select the best models for *MxRRI, MxRTI* and *MxINT* in terms of performance. At each simulation phase (movement), every reader prepares the entry to the models and predicts the presence of collision. Therefore, the reader adds the *ID* and *Position (x,y)*, then randomly selects a frequency and time slot and calculates the number of readers in RRI, RTI, and INT. The models can then predict the presence of collisions and take the appropriate action for *N* iterations according to Table 2.

**Table 2.** The prediction results and their corresponding actions.

| Model | Prediction | | | Action | | |
|---|---|---|---|---|---|---|
| | RRI | RTI | INT | New Freq | New TS | Tag Interrogation |
| MxRRI and MxRTI | 0 | 0 | - | × | × | √ |
| | 1 | 0 | - | √ | × | × |
| | - | 1 | - | √ | √ | × |
| MxINT | - | - | 0 | × | × | √ |
| | - | - | 1 | √ | √ | × |

### 3.4. Illustrative Example

To understand the process, Figure 6 illustrates an example using the RRI, RTI and INT learning models for the reader R1.

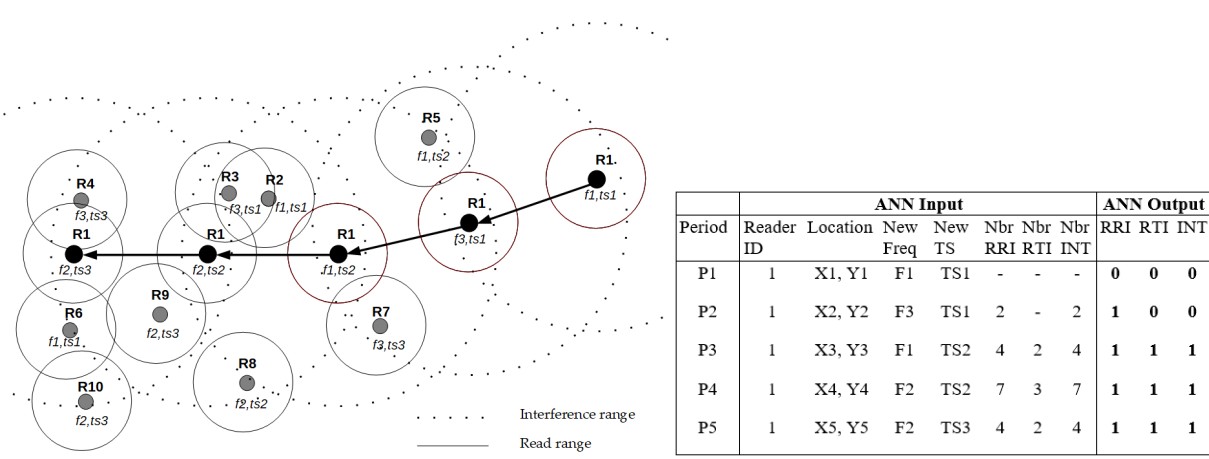

**Figure 6.** A scenario of a mobile reader in a RFID network.

After the collection and learning phase, reader R1 initiates the first test at period P1 by selecting data channel F1 and time slot TS1, and then calculates the number of readers that can be in RRI and RTI (Figure 6). Thus, as a result, the model does not detect any collision and starts tag interrogation.

In the next step (P2), reader R1 selects new frequency F3 and time slot TS1, and then detects two readers in RRI and RTI (Figure 6). The model result predicts the presence of RRI collision. At this point, the reader will attempt a new frequency for N iterations. In the remaining periods P3, P4 and P5, the reader model R1 predicts the presence of RRI and RTI collisions. Therefore, the reader tries new frequencies and time slots for N iterations.

In summary, the following contributions are achieved by our collision detection; in the first stage, we run different simulation scenarios for the RFID network. At each movement, the readers collect the current parameters (dataset input) and the corresponding collision result (dataset output) to build the dataset at the end of the simulation and share it with all readers in the network.

1. Each reader creates its own Neural Network object in order to insert the received dataset by the network readers;
2. Readers then train their Neural Networks to obtain the three required models (RRI, RTI and INT) that will be distributed to the other readers in the network;
3. Eventually, the readers choose and validate the optimal models to use for collision prediction.

## 4. Simulation and Results

### 4.1. Environment and Simulation Parameters

Considering a healthcare infrastructure as a base for our deployment simulation, we considered that our RFID network included sensor tags carried by patients and RFID readers by medical staff. We conducted MATLAB simulations for the reader mobility models (Figure 7) [51].

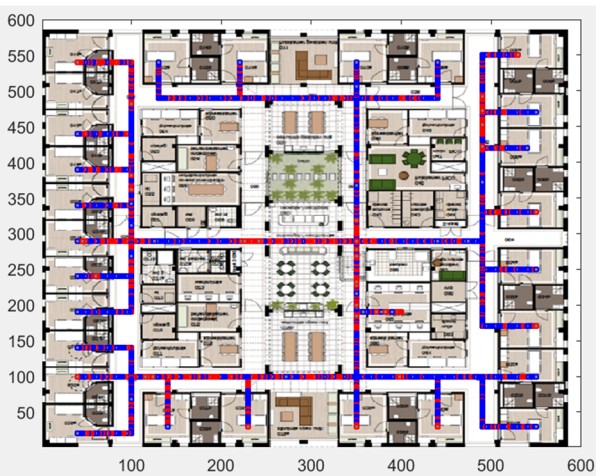

**Figure 7.** RFID reader network for directed mobility.

For this purpose, we created and implemented the following models and entities:

- Readers and tag RFID network entities;
- Reader–tag and reader–reader communication modules;
- RRI and RTI collision models;
- A learning and resource prediction algorithm;
- TDMA and FDMA for MAC layer in readers.

The goal was to evaluate our new anti-collision mechanism according to different criteria of mobility, resources and density.

The simulation parameters are provided in Table 3. The mobile readers are deployed so that all readers can communicate with each other through a control channel. Readers use the available data channels and each channel supports the time slot frame.

The input and output data for the Neural Network are presented as follows:

- Reader ID: scales to the interval [0–10];
- Location X and Y: scales to the interval [0–10];
- New Freq: [2,4,6,8,10];
- New TS: [2,4,6,8,10];
- Nbr RRI, Nbr RTI and Nbr INT: scales to the interval [0–10];
- RRI, RTI and INT: Boolean [0,1].

**Table 3.** Simulation parameters.

| Parameter | Value |
|---|---|
| Simulation range | $600 \times 600$ m |
| Number of readers | 10, 20, 30, 40, 50 |
| Number of tags | 1000 |
| Reader and tag position | Random |
| Type of antenna | Omni-directional |
| Read range of data channel (rr) | 3 m |
| Collision range of data channel (cr) | 562 m |
| Read range of control channel (crr) | 6 m |
| Number of data channel | 2, 4, 6, 8, 10 |
| Number of time slots | 2, 4, 6, 8, 10 |
| Number of control channels | 1 |
| Input layer neurons of all ANNs | 7 |
| Output layer neurons of all ANNs | 1 |
| Hidden layer number of all ANNs | 3 |
| Hidden layer neurons | 18 |
| Training method | Back propagation |
| Train function | trainlm |
| Maximum number of epochs to train | 1000 |
| Transfer function | Tansig, logsig |
| Number of periods | 1000 |
| Dataset size | nbr periods $\times$ nbr readers |
| Division of data for training | 70% |
| Division of data for validation | 15% |
| Division of data for testing | 15% |
| Performance function | Mean Square Error |

We used the *"trainlm"* training function based on *the Levenberg–Marquardt optimization* to update the weight and bias values. For such an RFID system, *trainlm* is usually the fastest and recommended back-propagation algorithm for supervised algorithms. It requires less memory than other algorithms [52]. Weight and bias properties are present in Appendix A.

We chose the following anti-collision protocols using the distributed communication principle (similar to our approach) to compare the performance: Pulse, CORA, MCMAC and FTSMAC.

Performance was evaluated according to two main criteria: *collision prediction* and *system performance.*

The system performance represents the throughput of readers as follows:

$$SystemPerformance(\%) = \frac{Total_{success} \times 100}{Total_{interrogation}} \tag{5}$$

where $Total_{success}$ is the number of successful tag interrogations and $Total_{interrogation}$ represents the total number of Tag interrogations.

The collision prediction, which allows us to evaluate the ability of the readers to avoid collisions, is presented as follows:

$$CollisionPrediction(\%) = TP + TN \tag{6}$$

where *TP* is a confusion matrix true positive and *TN* is a confusion matrix true negative of the ANN.

### 4.2. Performance and Tracking of Collision Predictions

In this section, we will study and implement the model obtained in the learning phase for collision prediction. Figure 7 tracks the prediction history for the movements (1000 movements for each reader) of 50 readers in the RFID using four frequency and four time slots. The red color identifies the position of the false prediction collision and the blue

color represents the true prediction. From this figure, we notice that the correct predictions dominate for the mobility models. The resulting ANN learning models are as follows: the successful prediction of the RTI collision type is 70% and that of the RRI collision type is 75%.

The best validation performance results for the Neural Network model described above using different datasets (datasets size : $50 \times 1000 = 50,000$ line) in Table 4 show the consistency and stability of the performance for different experiments. Figures 8 and 9 illustrate the performance of the RRI and RTI models for different datasets related to the deployment of readers using 10 frequencies and TSs. The results of the RRI model were similar at around 0.08, while they were different for RTI at between 0.04 and 0.11.

**Table 4.** Model performance for different datasets.

|           | RRI Model | RTI Model |
|-----------|-----------|-----------|
| Dataset 1 | 0.092179  | 0.10571   |
| Dataset 2 | 0.094731  | 0.10626   |
| Dataset 3 | 0.086352  | 0.10309   |
| Dataset 4 | 0.090341  | 0.10636   |
| Dataset 5 | 0.088141  | 0.10927   |
| Dataset 6 | 0.087115  | 0.10341   |
| Dataset 7 | 0.089247  | 0.10869   |

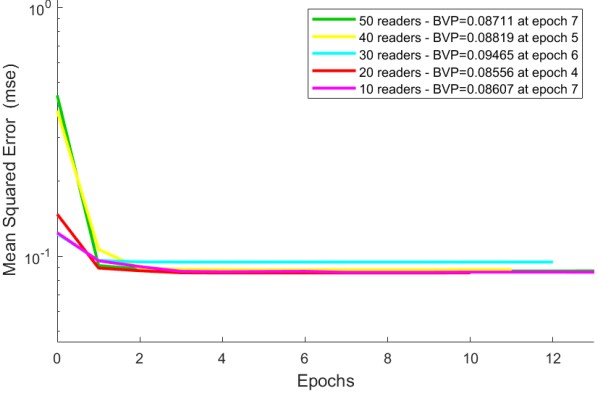

**Figure 8.** Best validation performance for RRI model.

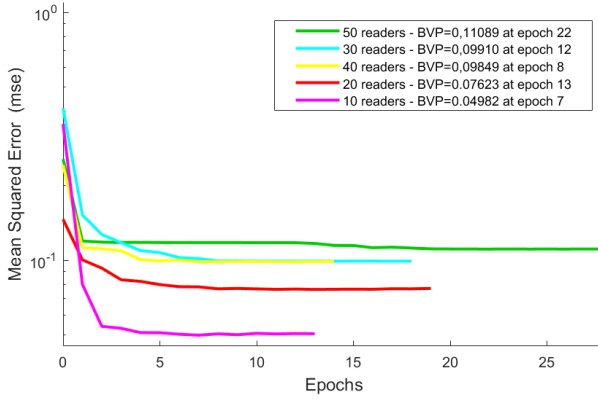

**Figure 9.** Best validation performance for RTI model.

### 4.3. Collision Prediction Evolution

In order to consolidate the choice of ANN model discussed in Section 3, we compared it performance in terms of collision prediction with Logistic Regression and the Decision Tree for 10 to 50 readers using 10 frequencies and time slots. Figures 10 and 11 show the RRI and RTI collision prediction results for different reader deployment scenarios. The RRI

prediction model ANN was most effective for 10, 20 and 40 readers. On the other hand, ANN and Logistic Regression were suitable for 30 and 50 readers. For the RTI model, ANN was most effective for 40 readers, while ANN and Logistic Regression could be used for 10, 20, 30 and 50 readers.

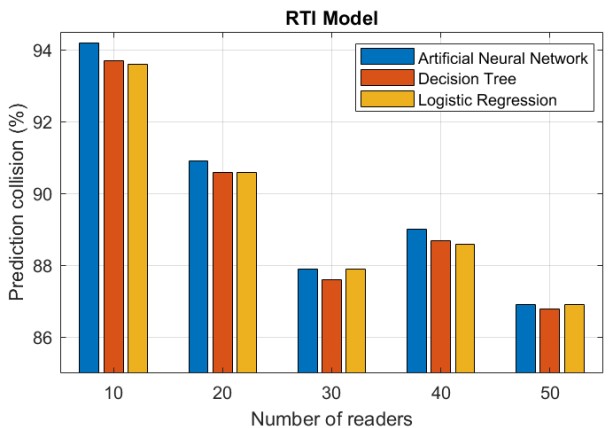

**Figure 10.** Performance of RRI collision prediction methods.

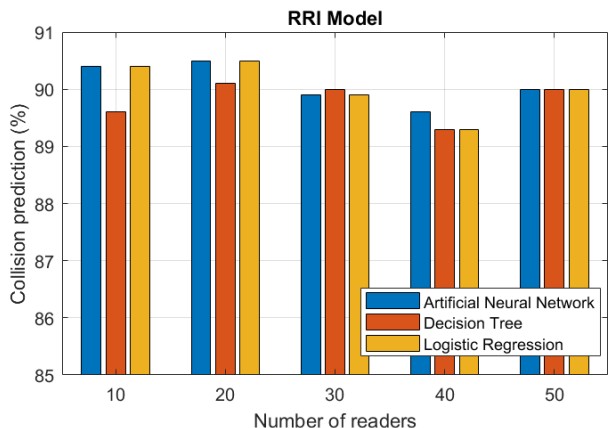

**Figure 11.** Performance of RTI collision prediction methods.

In summary, we can conclude that the ANN method was most efficient for our deployment model, hence our choice to use it for our prediction algorithm.

Figure 12 illustrates the evolution of collision prediction for the mobility model as a function of the number of readers, and the frequency and the time slot used by each reader. We noticed that RTI prediction model outperformed the other models for a dense network (40 and 50 readers), while in a medium network (30 readers) the RRI and RTI models were similar. Finally, in a sparse network (10 and 20 readers), the RRI model outperformed the RTI model.

### 4.4. Collision Prediction and System Performance

In Figures 13 and 14, we compare the performance of the different models based on the number of frequencies (Figure 14) and the number of time slots (Figure 13). In Figure 13, the RTI prediction model is efficient and stable based on the number of available TSs. In Figure 14, the RRI and RTI models are almost similar with an advantage for the RRI model using low frequencies (2 to 4 frequencies) and for the RTI model with high frequencies (8 to 10 frequencies).

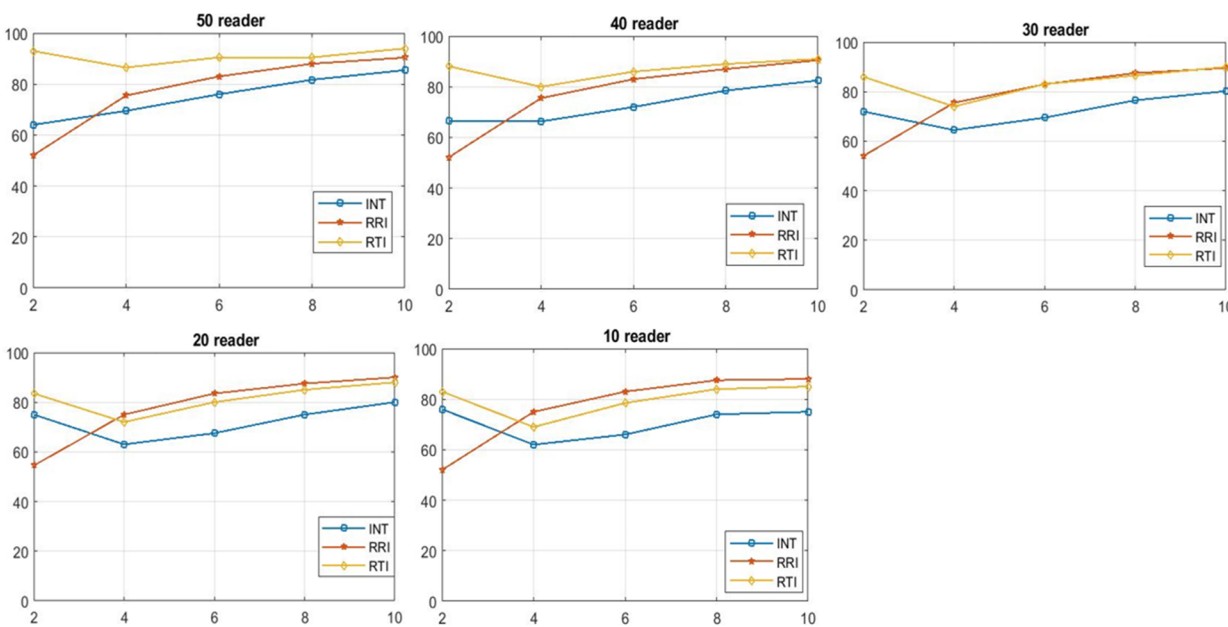

**Figure 12.** Collision prediction vs. number of used resources (frequency and time slots) for the directed mobility model.

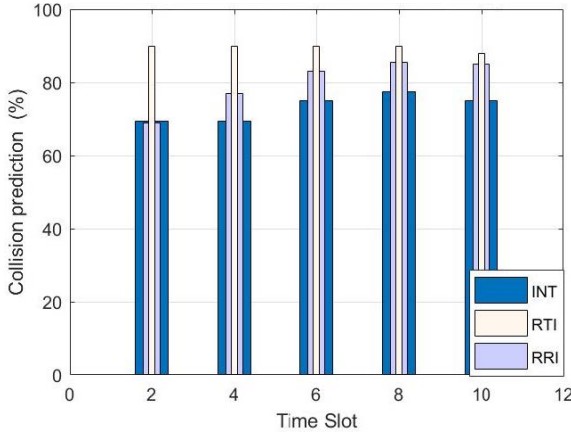

**Figure 13.** Collision prediction vs. number time slots (50 readers, 10 frequencies).

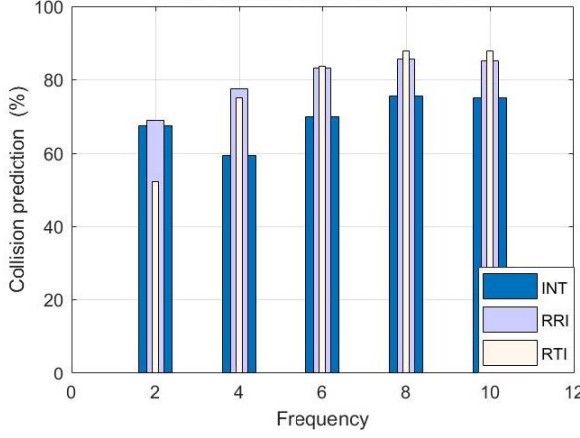

**Figure 14.** Collision prediction vs. number of frequencies (50 readers, 10 time slots).

In Figure 15, we compare the read rate and coverage of our protocol using different prediction models from the literature such as Pulse, MCMAC, CORA and FTSMAC. This

comparison is performed for a deployment of 10 to 50 readers using three frequencies and time slots. The simulation showed that our protocol achieved the highest results independently of the number of RFID readers. It reached its maximum for a network of 10 to 20 readers. The results confirmed the efficiency of this approach by using two prediction systems of each collision type. For the other algorithms, FTSMAC used Scheme-FTSMAC for resource reuse, which was suitable for dense networks but not for the sparse networks, while MCMAC was less efficient because it suffered from RTI problems due to simultaneous communication using different frequencies.

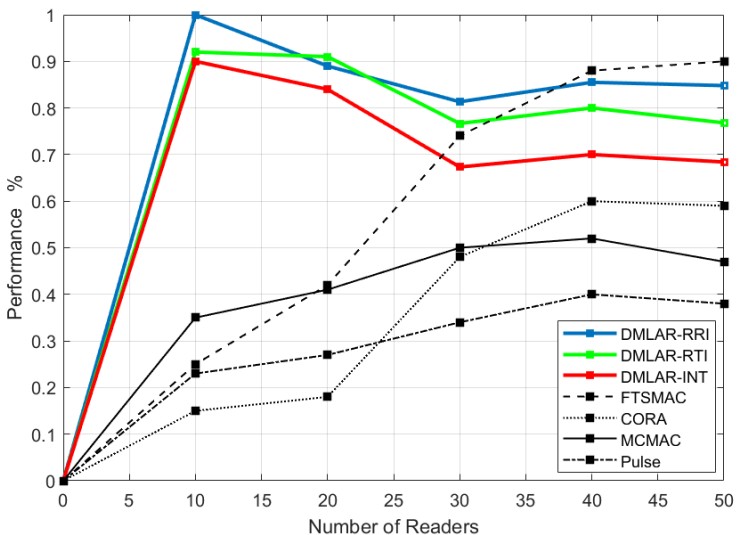

**Figure 15.** Performance vs. number of readers.

*4.5. Cost Comparison (Using 10 Frequencies and TSs)*

4.5.1. Failed Interrogations

In healthcare, collected data from patients are extremely important and critical. Figure 16 illustrates the number of failed interrogations as a function of the number of deployed readers. The CORA and MCMAC protocols followed a similar evolution of the number of failed requests. Since CORA does not support further resources, MCMAC covered the RRI collision problem.

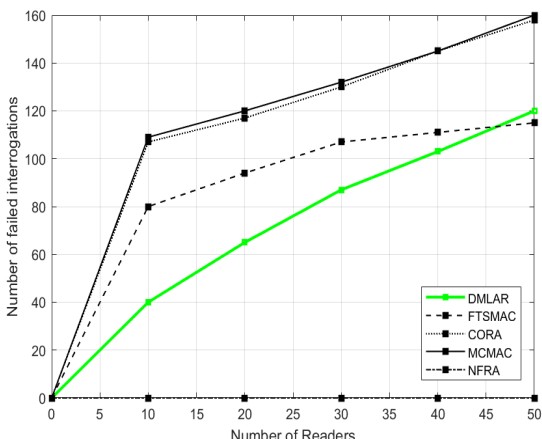

**Figure 16.** Failed interrogations vs. number of readers.

4.5.2. Network Overload

The network overload represents the number of control packets exchanged between the readers. We note that our DMLAR method presented the lowest rate of overload, as

shown in Figure 17, while this parameter increased for MCMAC. Furthermore, NFRA functioned as a server to broadcast the control commands to the readers, which increased the number of bytes exchanged.

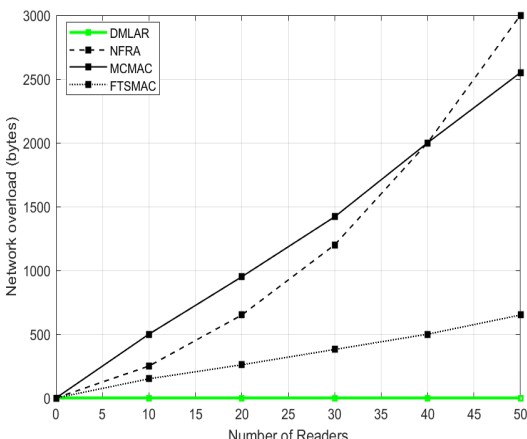

**Figure 17.** Network overload vs. number of readers.

### 4.5.3. Interrogation Delay

The overall reading delay for the readers was determined as the time necessary for the reader to interrogate all the tags. Figure 18 shows that NFRA consumed more time due to the reader–server communication. FTSMAC and MCMAC required a reasonable delay using advanced notification systems, whereas the DMLAR approach required a smaller delay since it used this delay to predict collisions in the ANN models instead of using a notification system.

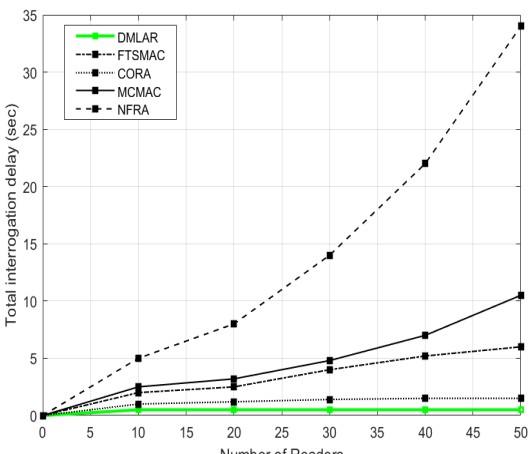

**Figure 18.** Total interrogation time vs. number of readers.

### 4.5.4. Energy Consumption

Energy efficiency is specified as the total energy required by the Reader-to-Tag Interrogation. As shown in Figure 19, NFRA had high energy consumption since it used a central server to manage all interactions. MCMAC and FTSMAC followed a parallel development with a difference of almost 15 w using 40 and 50 readers. On the other hand, the DMLAR consumed minimum energy due to the absence of a notification system; these consume more energy due to their communication modules.

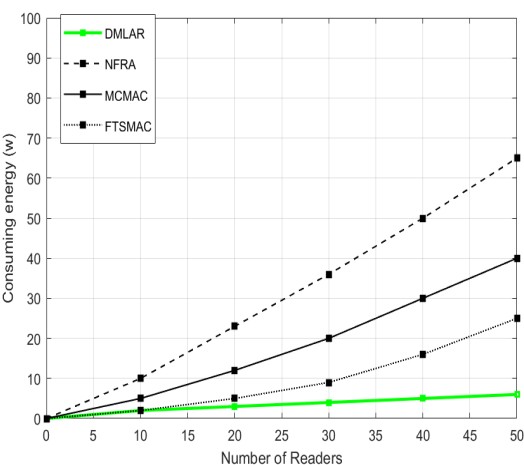

**Figure 19.** Consuming energy vs. number of readers.

## 5. Conclusions and Future Work

This paper proposes a novel anti-collision algorithm for the frequency and time slot resource management of RFID readers in mobile environments. This protocol allows readers to individually control their resources according to the selected collision prediction model of RFID network readers after a learning phase. In our approach, readers use a collision prediction model for each type of interference—Reader-to-Reader Interference and Reader-to-Tag Interference—for greater accuracy and correct prediction.

The dataset used for learning is updated with each movement by which the readers change position and resources are used. At the end of the simulation, all readers broadcast their dataset to obtain a larger database.

The goal of this proposal is to allow as many readers as possible to interrogate tags without collision in order to increase the performance of mobile RFID networks in an autonomous and smarter way.

The simulation results show the ability of this algorithm to detect RRI and RTI collisions and therefore obtain more successful interrogations for different mobility modes. In addition, the number of available readers, frequencies and time slots does not affect the performance of our protocol. Additionally, according to the different cost types, we eventually approved the compatibility of this algorithm for RFID networks.

As a future work, we will work to improve the performance of this learning system for high-density RFID networks. To achieve this, we will improve our distributed notification system to support the large number of datasets provided by the readers. Moreover, we will use the time constraint as an input in the ANN model to control the time movement of the readers. In the application phase of our algorithm, we will integrate an Artificial Immune Network for the resource allocation process after collision prediction using the ANN.

**Author Contributions:** The authors confirm contributions to the paper as follows: study conception and design, R.M.; programming, R.M.; analysis and interpretation of results, R.M., M.O.; data curation, R.M.; draft manuscript preparation, R.M., M.O., N.I.; review and editing, H.S., R.M.; reviewing the results and confirming the final version of the paper, R.M., M.O., N.I.; project administration, K.M. All authors have read and agreed to the published version of the manuscript.

**Funding:** This research received no external funding.

**Institutional Review Board Statement:** Not applicable.

**Informed Consent Statement:** Not applicable.

**Data Availability Statement:** Not applicable.

**Conflicts of Interest:** The authors declare no conflict of interest.

## Appendix A

| Layer | Weights |
|---|---|
| Input weights—net.IW{1} | 0.0044 −0.2514 −0.0070 −0.0048 0.1636 0.3204 −0.2330<br>−0.0057 0.1024 −0.0001 −0.0029 0.2460 −0.5623 0.4196<br>−0.0001 −0.0156 0.0013 −0.0000 −0.0041 −0.0594 −0.0037<br>0.0001 0.0017 0.0009 −0.0016 2.8498 0.1155 0.0402<br>−0.0002 −0.0059 0.0011 −0.0003 0.0264 −0.0361 −0.0008<br>0.0005 0.0055 0.0009 0.0001 −0.7914 −0.0932 −0.3751<br>0.0115 −0.1119 0.0055 0.0003 0.4235 −0.1432 −0.2438 |
| Layer weights—net.LW{2,1} | 0.8165 −0.5806 −0.9095 −0.6005 0.5392 0.1841 0.2697<br>−0.2166 −0.3655 0.4676 0.2382 −1.4459 0.5032 −0.0841<br>−1.4496 −0.2913 0.7685 −0.5432 0.2806 −0.5419 −0.4694<br>0.1064 −0.4785 −0.1471 0.0264 0.1680 1.0828 0.1746<br>−0.1687 0.9781 −1.1379 −0.5375 0.6862 0.4651 0.8598<br>−0.3313 0.4047 0.7772 1.6814 −1.5036 0.1976 0.6137<br>−0.9726 0.4199 0.3512 0.7845 0.7522 0.5266 −0.6154<br>−0.4808 −0.3502 0.6576 0.3655 0.0776 0.6288 −0.6563<br>−0.0556 0.4930 −0.6053 1.0856 −0.5258 0.7849 0.2940<br>−0.2372 −0.3053 −0.7638 −0.1561 −0.1034 −0.7218 0.5194 |
| Layer weights—net.LW{3,2} | −0.5501 0.8311 −0.9173 1.4929 −0.8609 1.5320 0.3525 0.5138<br>1.2497 −0.8823 |

| Layer | Bias Vector |
|---|---|
| Biases—net.b{1} | 1.2809<br>3.1066<br>0.1546<br>−1.5453<br>1.0101<br>−0.5471<br>1.3648 |

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
