# Peer review of "DMLAR: Distributed Machine Learning-Based Anti-Collision Algorithm for RFID Readers in the Internet of Things"

_computers, doi:10.3390/computers11070107_

Round 1

Reviewer 1 Report

The authors present in this paper proposes a novel anti-collision algorithm for frequency and time slot resource management of RFID readers in mobile environments. The article is very well written can be considered an example to follow. This presented article has all the necessary elements for a professionally developed work: protocol analysis, simulation of anti-collision algorithm and results. The comparative study (Figure 11) with other algorithm from the literature is well documented and gives readers a clear picture of the progress made in this paper. The conclusions are fully supported by the simulation results and comparative study. Based on the elements presented, I believe that this article can be published in this form.

Author Response

Dear reviewer,

We would like to thank you very much; 

Yours sincerely,

Reviewer 2 Report

This paper presents an anti-collision protocol based on feed-forward Artificial Neural Network methodology for distributed learning between RFID readers to predict collisions and ensure an efficient resource allocation by considering the mobility of tags and readers.

Some details on the neural network input data should be given. Are these data pre-processed, e.g. scaled, before they are used in neural network training?

There is no mention of training data and testing data. Do you use all the data for neural network training? If so, there can be over-fitting which results in poor generalisation performance. The neural network performance on a set of new data (unseen data) should be given to demonstrate the generalisation performance.

Reviewer 3 Report

The paper proposes to apply feed-forward multi-layer perceptrons to classify collision types in RFID readers. The simulation shows promising results, but I think the paper can be strengthened by addressing the following points:

  1. It's unclear to me why the authors chose to use ANN in the first place. Would a more straightforward model like logistic regression or boosted tree not work? Explaining the choice in-depth or adding a baseline using logistic regression to the experiment would be appropriate. 
  2. I'm skeptical about the robustness of any learning-based system. Will the proposed approaches learn the pattern shift (if there is any) over time? The authors should at least discuss it or leave it as future work.
  3. Many prior works are mentioned in Section 2, but they are not properly compared in the evaluation Section. The authors should justify why some algorithms are compared, and some are not. Or add more experiments. 

Round 2

Reviewer 2 Report

I am satisfied the revised manuscript which can be accepted.

Author Response

We would like to thank you for your acceptance.

Reviewer 3 Report

I'm not convinced by the authors' response to my point 1. I still think it's worth running a baseline experiment using simpler ML models. Otherwise, using ANN might just be overkill here. I understand it might be a lot more work, so I'm leaving it up to the authors. If additional experiments are hard, please at least add more justifications for the usage of ANN. 
